# Bioinspired Dopamine and *N*-Oxide-Based Zwitterionic Polymer Brushes for Fouling Resistance Surfaces

**DOI:** 10.3390/polym16121634

**Published:** 2024-06-09

**Authors:** Zhen Zhou, Qinghong Shi

**Affiliations:** 1Department of Biochemical Engineering, School of Chemical Engineering and Technology, Tianjin University, Tianjin 300350, China; zhou_zhen@tju.edu.cn; 2Key Laboratory of Systems Bioengineering (Ministry of Education), Tianjin University, Tianjin 300072, China

**Keywords:** antifouling, zwitterionic polymer, *N*-oxide, surface modification, adsorption, BSA, QCM-D

## Abstract

Biofouling is a great challenge for engineering material in medical-, marine-, and pharmaceutical-related applications. In this study, a novel trimethylamine *N*-oxide (TMAO)–analog monomer, 3-(2-methylacrylamido)-*N*,*N*-dimethylpropylamine *N*-oxide (MADMPAO), was synthesized and applied for the grafting of poly(MADMPAO) (*p*MPAO) brushes on quartz crystal microbalance (QCM) chips by the combination of bio-inspired poly-dopamine (*p*DA) and surface-initiated atom transfer radical polymerization technology. The result of ion adsorption exhibited that a sequential *p*DA and *p*MPAO arrangement from the chip surface had different characteristics from a simple *p*DA layer. Ion adsorption on *p*MPAO-grafted chips was greatly inhibited at low salt concentrations of 1 and 10 mmol/L due to strong surface hydration in the presence of charged N^+^ and O^−^ of zwitterionic *p*MPAO brushes on the outer layer on the chip surface, well known as the “anti-polyelectrolyte” effect. During BSA adsorption, *p*MPAO grafting also led to a marked decrease in frequency shift, indicating great inhibition of protein adsorption. It was attributed to weaker BSA-*p*MPAO interaction. In this study, the Au@*p*DA-4-*p*MPAO chip with the highest coating concentration of DA kept stable dissipation in BSA adsorption, signifying that the chip had a good antifouling property. The research provided a novel monomer for zwitterionic polymer and demonstrated the potential of *p*MPAO brushes in the development and modification of antifouling materials.

## 1. Introduction

Biofouling has increasingly become one of the critical problems for the design, development and application of industrial and medical facilities including implantable medical devices [1,2], membranes [3,4], biosensors [5,6], and marine sailing [7,8,9], etc. As is known to all, a typical process of biofouling is always initialized by protein binding at solid–liquid interfaces, followed by the adhesion of cells or microorganisms. Unfortunately, it leads to a dramatic decrease in the performance and lifetime of the facilities, and even a serious threat to human lives in the medical industry [10]. Therefore, the design and development of low-binding and even non-fouling interfaces are of more importance for those facilities in industrial, marine, and medical applications.

Among all the approaches, interface modification has become a core subject of academic and engineering research to minimize and even eliminate fouling. Up to now, diverse polymers have been used for interface modification, and polyzwitterions are deemed to be excellent materials for antifouling coating design to solve the problem of bio-pollution [11]. Why do polyzwitterions have excellent antifouling ability? Firstly, polyzwitterions, which possess an equal number of cationic groups and anionic groups [12], can bind water molecules strongly via electrostatically induced hydration [13]. The tightly bound water molecules can hinder protein and cell adsorption by increasing the energy barrier [14,15]. What is more, zwitterionic polymer brushes in water tend to be in the “collapse state” and have hydrophobic surface properties due to the electric interaction of zwitterionic units in one polymer chain or between polymer chains (inter/intra interactions). In salt solutions, the brushes stretch due to the presence of ion screens in the inter/intra interactions, leading to surface hydrophilicity [16,17,18,19]. It is called anti-polyelectrolyte behavior. It means that zwitterionic polymer brushes adsorb biomolecules/cells at low salt concentrations and desorb at high salt concentrations. However, three major classes of polyzwitterions, including poly(carboxybetaine) (PCB) [11,20], poly(sulfobetaine) (PSB) [8,21], and poly(phosphocholine) (PPC) [15,22], have distinct conformational changes with ionic strength increasing [23]. Recently, a new class of ultralow fouling biomaterials, trimethylamine *N*-oxide (TMAO)-derived zwitterionic polymers (*p*TMAO), was developed by Jiang’s group [10]. The result demonstrated the superior hydrophilicity and non-fouling properties of *p*TMAO, and the *p*TMAO-coated surface exhibited ultralow protein adsorption and minimal immunogenicity. Furthermore, Jiang’s group illustrated the surface hydration of *p*TMAO and the effects of salts and proteins on such surface hydration, and the surface hydration can only be moderately reduced by highly concentrated salt solutions, owing to the shorter distance between the positively and negatively charged groups than other zwitterionic polymers [24]. Zhang et al. reported that *p*TMAO was synthesized by in situ oxidation of block copolymers, (2-dimethylaminoethyl methacrylate)-block-polystyrene (PDMAEMA-*b*-PS), and the *p*TMAO-coated membranes exhibit strongly hydrophilicity and flexibly tunable perm-selectivity [25]. However, the fourth class of non-fouling zwitterionic materials, *p*TMAO, has received little attention from researchers.

Mussel-inspired chemistry is the universal bridge between the *p*TMAO brush and a substrate [26]. By simply dipping, polydopamine can adhere to the surface of organic and inorganic materials, including titanium [15], silicon nanoparticles [27], and membranes [4], etc. Then, a strategy was put forward to realize a highly uniform and enhanced stable PDA coating by using CuSO4/H_2_O_2_ as a trigger [28]. The PDA coating contains amino groups and phenolic hydroxyl groups [2], which can serve as the anchor point for secondary surface modification [29]. Then, two methods, “grafting to” and “grafting from”, can be used to attach the *p*TMAO brush onto the surface of the PDA coating. The “grafting from” method via surface-initiated atom transfer radical polymerization (SI-ATRP) enables the production of coatings with high packing densities [13]. According to the works of literature we found, it has not been reported that the combination of PDA and *p*TMAO has been used for surface modification of substrates.

The quartz crystal microbalance with dissipation (QCM-D) technique is one of the outstanding methods to investigate the conformational change of *p*TMAO during ion/protein adsorption [30]. By measuring frequency (Δ*f*) and energy dissipation shift (Δ*D*), QCM-D can analyze mass and viscoelasticity change simultaneously on sensor surfaces [31]. Numerous researchers have used the adsorption mass as an important indicator for evaluating the anti-protein adsorption ability of materials. Usually, QCM-D is a technique for quantitative calculation of adsorption mass by the Sauerbrey equation [32]. In the past few years, the QCM-D data have been further analyzed and combined with other characterization data. As reported by Kushiro et al. [33], QCM-D was utilized to quantitatively monitor the protein adsorption processes on various self-assembled monolayer surfaces. In the study, the Δ*D*/Δ*f* value (which is the slope of the *D*-*f* plots) was used to characterize material–protein interactions and material–protein–cell interactions can be described with the characteristic features of QCM-D *D*-*f* plots. Recently, You et al. depicted the adsorption behavior between proteins and polyelectrolyte brushes with *D*-*f* plots [34].

Overall, in this study, an original zwitterionic monomer belonging to the family of organic osmolytes, 3-(2-methylacrylamido)-*N*,*N*-dimethylpropylamine *N*-oxide (MADMPAO as shown in Figure 1a), was synthesized by oxidizing *N*-(3-dimethyl-aminopropyl) methacrylamide (DMAPMA). As shown in Figure 1b, a poly(MADMPAO) (*p*MPAO) brush was grafted onto a QCM chip in a combination of mussel-inspired chemistry and SI-ATRP technique to develop the *p*MPAO-grafted chips. The chips were well characterized with the help of various techniques in the field of morphology, hydrophilicity, elemental composition, and chemical structure by contact angle instrument, atomic force microscopy (AFM), and X-ray photoelectron spectroscopy (XPS) measurement. Lastly, this investigation utilized real-time data from QCM-D to analyze the adsorption behavior of salt (NaCl, Na_2_SO_4_, and MgCl_2_) and bovine serum albumin (BSA) onto the sensors’ surface. To the best of our knowledge, it is the first time, in the literature, that the ions and protein adsorption behavior on *p*DA-*p*TMAO brushes are investigated by QCM-D.

## 2. Materials and Methods

### 2.1. Materials

Dopamine hydrochloride (purity: 98%) and bovine serum albumin (BSA; purity: ≥98%) were obtained from Sigma-Aldrich (St. Louis, MO, USA). Diethylenetriaminepentaacetic acid (purity: 98%) and DMAPMA (purity: 98%) were received from Tianjin Heowns Biochem LLC (Tianjin, China). Tris(2-dimethylaminoethyl)amine (Me_6_TREN; purity: 99%) were purchased from J&K Chemical (Beijing, China) and hydrogen peroxide (H_2_O_2_; purity: ≥30%) was obtained from Titan Scientific Co., Ltd. (Shanghai, China). Copper(I) bromide (CuBr; purity: 99.5%) was supplied by Dingguo Changsheng Biotechnology Co., Ltd. (Beijing, China). Bare gold QCM chips (QSX301, 14 mm diameter, 5 MHz) were obtained from Jiaxing Jingkong electronic Co., Ltd. (Jiaxing, China) and used for QCM-D experiments. Ultrapure water, with a specific resistivity of 18 MΩ·cm, was used to produce all the solutions. Other reagents were analytical grade from local suppliers.

### 2.2. Synthesis of MADMPAO

An inventive TMAO-derived monomer, MADMPAO, was synthesized as described in a previous publication with some modifications [10]. Ultrapure water (30 mL) and diethylenetriaminepentaacetic acid (80 mg) were added to a three-necked flask and stirred mechanically until the white powder dissolved completely. After oxygen gas was slowly purged into the solution, the flask was heated to 60 °C, and hydrogen peroxide (30% solution, 10 mL) was slowly added. Then, *N*-(3-Dimethylaminopropyl) methacrylamide (14.47 g) in 10 mL of ultrapure water was added dropwise for 30 min. The reaction was carried out for 6 h at 60 °C. After the reaction, water was removed by rotary evaporation at 80 °C. The final product is a colorless and viscous liquid. The product was characterized with an Agilent 6550 Q-TOF spectrometer (Agilent Technologies Co., Ltd., Santa Clara, CA, USA) in a positive ion mode. The flow rate of the sample was 0.3 mL/min, and the temperature and flow rate of desolvation gas were 150 °C and 15 L/min, respectively. High-resolution mass spectrometry (HRMS) analysis was carried out by using an Agilent 6550 Q-TOF spectrometer. ^1^H-NMR spectrum was recorded on a Varian Unity Inova 500 MHz nuclear magnetic resonance spectrometer (Bruker Corporation, Billerica, MA, USA) to conform the product. ^1^H NMR (500 MHz, D_2_O): Δ = 5.65 (q, J = 1.0 Hz, 1H), 5.42 to 5.38 (m, 1H), 3.30 (td, J = 8.4, 7.6, 4.1 Hz, 4H), 3.13 (s, 6H), 2.06 to 1.96 (m, 2H), 1.89 to 1.85 (m, 3H).

### 2.3. Surface Coating on QCM Sensors by Polydopamine

Prior to *p*DA coating, the bare gold QCM chip was activated by fresh piranha solution for 10 min and then rinsed with ultrapure water and absolute ethanol 3 times. The activated chip was immersed into 2–4 mg/mL DA, 5 mmol/L CuSO_4_ and 19.6 mmol/L H_2_O_2_ in 50 mmol/L Tris-HCl buffer (pH 8.5) [35]. *p*DA coating was carried out in a water bath at 25 °C and 90 rpm for 40 min. After that, the *p*DA-coated chips were washed with ultrapure water and absolute ethanol 3 times and then dried in a 60 °C oven overnight for the polymer grafting. The chips were named Au@*p*DA-2 at 2 mg/mL DA, Au@*p*DA-3 at 3 mg/mL DA, and Au@*p*DA-4 at 4 mg/mL DA.

### 2.4. Polymer Grafting on Au@pDA

Polymer grafting was initiated by the coupling ATRP initiator on a *p*DA-coated chip via nucleophilic substitution. In brief, 10.0 mL tetrahydrofuran (THF) and 1.2 mL triethylamine (5.0 mmol) were added into a three-necked flask cooled in an ice-water bath, and the mixture was stirred for 20 min. Then, the *p*DA-coated chip was transferred, gold surface up, into the flask. The chip was immersed for 10 min, and then 2-bromoisobutyryl bromide (BiBB) (1.175 g, 5.0 mmol) in 3 mL THF was added dropwise in 5 min [15]. After incubation in an ice-water bath for 40 min, the mixture was reacted for about 12 h at 25 °C. The product was rinsed in ethanol and ultrapure water and then dried for polymer grafting. The initiator-coupled chips were named Au@*p*DA-2-Br, Au@*p*DA-3-Br, and Au@*p*DA-4-Br, respectively.

*p*MPAO was grafted on the initiator-coupled chips by SI-ATRP. In brief, MADMPAO monomer (1.86 g, 10 mmol), water (0.4 mL), and methanol (3.6 mL) were added into a 25 mL Erlenmeyer flask, and nitrogen was blown to deoxygenation for 20 min. Then, CuBr (14.35 mg, 0.1 mmol) and Me_6_TREN (23 mg, 0.1 mmol) were added into the flask, and it was deoxygenated continuously. The chip was transferred, initiator-coupled surface up, into the flask immediately. After being purged with nitrogen for 20 min, the flask was sealed and incubated overnight in a shaking bath at 25 °C and 120 rpm. The products were washed with alternating ethanol and ultrapure water to remove residual reactants and then dried for adsorption study. The *p*MPAO-grafted chips were named Au@*p*DA-2-*p*MPAO, Au@*p*DA-3-*p*MPAO, and Au@*p*DA-4-*p*MPAO, respectively.

### 2.5. Characterization

In this study, the surface morphology of bare gold, *p*DA-coated, and *p*MPAO-grafted chips was characterized with CSPM5500 AFM (Guangzhou, China). Water contact angles were measured with JC2000D1 contact angle system (Zhongchen Digital Technology Equipment Co., Ltd., Shanghai, China) after a 5 μL drop volume of pure water was placed on the sample surface. The measurement was repeated three times at different positions. XPS was employed to analyze the elemental composition of chips by Thermo Fisher K-Alpha+ X-ray photoelectron spectroscopy (Thermo Fisher Scientific Inc., Grand Island, NY, USA) with an aluminum monochromatic source and the photoelectron take-off angle of 90°.

### 2.6. Ion and BSA Adsorption Study by QCM-D

Ion adsorption and protein binding on *p*DA-coated and *p*MPAO-grafted chips were conducted with the QCM-D E1 instrument (Q-Sense, Gothenburg, Sweden) as described previously [23,36]. In brief, the chips were equipped in the QCM-D chamber, and the temperature was set to 25 ± 0.1 °C. Ultrapure water was pumped into the chamber at a flow rate of 100 μL/min until the baselines were stable. Salt solutions with different concentrations (1, 10, and 100 mmol/L) were pumped successively into the chamber for 10–20 min each. Then, ultrapure water was pumped into the chamber again until the stable baselines were achieved. In this experiment, NaCl, Na_2_SO_4_ and MgCl_2_ were applied to evaluate ion influence on polymer-grafted layers. All the frequency and dissipation measurements were taken from seventh overtone (*n* = 7). All QCM-D measurements were repeated at least three times.

BSA adsorption to *p*MPAO-grafted chips was monitored by the QCM-D at 37 ± 0.1 °C. In the experiment, 150 mmol/L NaCl in 200 mmol/L phosphate buffer (pH 7.4) was used as adsorption buffer, and BSA solution (100 μg/mL) was prepared in adsorption buffer. After the baselines were stable, the BSA solution was pumped for 30 min. Then, the adsorption buffer was injected until the baselines were achieved again. The adsorbed amount of BSA onto the sensors was calculated using the following Sauerbrey equation:(1)$$\Delta m=-C\Delta f_{n}/n$$
where *C* was the mass sensitivity constant and 17.7 ng/(cm·Hz) for a 5 MHz quartz crystal, *n* and Δ*f* were overtone numbers and their corresponding resonant frequency, respectively. In this study, the third, fifth, and seventh overtones were chosen for the calculation of the adsorbed amount by the Sauerbrey equation and the average adsorbed amount was reported.

## 3. Results

### 3.1. Synthesis of TMAO-Analog Monomer

In this study, an inventive TMAO-derived monomer, MADMPAO, was synthesized by oxidizing DMAPMA as shown in Figure 1a. Using 30% hydrogen peroxide as the oxidant, DMAPMA was completely converted in the presence of oxygen gas. Compared with the organic solvent extraction of TMAO monomer reported previously [37], a simpler green method, rotary evaporation, was adopted in this study for the monomer purification, and a colorless and viscous product was obtained. The NMR result is shown in Figure 2. In the ^1^H NMR spectrum of DMAPMA, the signals of methylene protons adjacent to nitrogen (3), methyl protons adjacent to nitrogen (4), and ester methylene protons (6) appeared at 2.31, 2.14 and 1.66 ppm, respectively. After the oxidation reaction, new peaks of the corresponding groups arose at 3.30, 3.13 and 2.01 ppm, respectively. These signals shifted to lower fields, indicating that there was a decrease in electron cloud densities of the corresponding protons under the inductive effect of N^+^ as an electron-withdrawing group [25]. HRMS analysis is shown in Appendix A of Appendix A. HRMS (mass/charge ratio): calculated for C_9_H_18_N_2_O_2_, 187.1368 ([M+H]^+^); found 187.1442. NMR and HRMS results confirmed a successful synthesis of MADMPAO.

### 3.2. Surface Modification and Characterization

In this study, an original *N*-oxide-based zwitterionic polymer was grafted on a QCM chip using MADMPAO as the monomer. Prior to the grafting, the bare gold chip was modified by coating with *p*DA [22]. As shown in Figure 1b, a uniform *p*DA-coated layer was formed on the bare gold chip by self-polymerization in the presence of CuSO_4_/H_2_O_2_. *p*DA possessed rich functional groups (e.g., amine, imine and catechol) for the initiator coupling [38]. To evaluate the effect of the *p*DA layer, 2, 3 and 4 mg/mL DA solutions were applied in *p*DA coating. Figure 3 shows AFM images of bare gold, *p*DA-coated, initiator-coupled, and *p*MPAO-grafted chips. It was seen in Figure 3a that there was a uniform and smooth surface on the bare gold chip characterized by a root-mean-square (rms) roughness of ~6.13 nm. Surface coating by *p*DA led to a marked increase in roughness (14.2–24.0 nm) and crowded small protuberances were observed on chip surfaces in Figure 3b. Furthermore, the roughness of the *p*DA layer decreased with an increase in initial DA concentration. It was also observed in 0.5 h DA coating on TiO_2_ substrates by Ding et al. [39]. A successful *p*DA coating is also validated by a marked decrease in water contact angles on *p*DA-coated chips in Appendix A of the Appendix A. As shown in Figure 3c, furthermore, smaller roughness ranging from 10.9 to 12.9 nm was found on the surface of initiator-coupled chips, revealing a smoother and stable *p*DA-coated layer on the initiator-coupled chip. It may be related to the introduction of an alkyl group in the initiator, inducing the layer collapse via intra-layer hydrophobic interaction. Accordingly, higher contact angles of the initiator-coupled chip (62.5–79.5°) were obtained in Appendix A of the Appendix A due to the presence of hydrophobic alkyl groups. After polymer grafting using the zwitterionic monomer, MADMPAO, via ATRP, *p*MPAO-grafted chips exhibited rougher surfaces than initiator-coupled chips in Figure 3d, especially on *p*MPAO-grafted chips with higher DA concentrations. Because zwitterionic heads contain both positive and negative charges, they underwent charge pairing in intra- and inter-polymer chains and induced the collapse of *p*MPAO brushes [40]. Therefore, *p*MPAO grafting only brought about a slight increase in rms roughness on *p*MPAO-grafted chips (13.4–18.8 nm) compared with initiator-coupled chips. Meanwhile, contact angles decreased to 42.8° on the Au@*p*DA-2-*p*MPAO chip, 48.4° on the Au@*p*DA-3-*p*MPAO chip, and 46.4° on the Au@*p*DA-4-*p*MPAO chip in Appendix A of the Appendix A. Recently, a similar result was obtained by Feng et al. in contact angle measurement of TMAO analog-modified polyamide membrane [41]. At the similar material hydrophilicity, furthermore, the higher water contact angle of the Au@*p*DA-3-*p*MPAO chip was attributed to the smaller roughness of the surface [42]. The morphological feature of chips demonstrated a successful grafting on the chip surface.

In this study, the surface elemental composition of representative Au@*p*DA-3, Au@*p*DA-3-Br, and Au@*p*DA-3-*p*MPAO chips is listed in Table 1. It was seen that the ratio of C/N and O/N were ~8.5 and ~2.1 on the surface of the Au@*p*DA-3 chip. The result was close to the theoretical C/N and O/N ratio of dopamine (C_8_H_11_NO_2_, C/N = 8, and O/N = 2) [43,44]. The initiator coupling led to a slight decrease in C atomic content from 73.44% to 69.8% and N atomic content from 8.65% to 7.9% on the chip surface while O atomic content increased from 17.91% to 18.61%. More importantly, characteristic peaks of bromine (Br3p and Br3d) were observed on the Au@*p*DA-3-Br chip in Figure 4a, demonstrating a successful coupling of BiBB to *p*DA-coated chips. After *p*MPAO grafting, atomic contents of C, N, and O increased to 72.61%, 8.05%, and 19.34%. The variety of elemental composition on the chip surface indicated the synthetic trajectory of *p*MPAO-grafted chips in Figure 1.

High-resolution XPS C1s and N1s spectra were further analyzed to gain an insight into the chemical functional groups in *p*DA-coated, initiator-coupled, and *p*MPAO-grafted layers on chip surfaces. As shown in Figure 4b, C1s spectra comprised three characteristic peaks at 284.68 eV (C=C, C-C, and C-H of phenyl), 286 eV (C-N, C=N, and C-OH), and 288.3 eV (O=C-O or O=C-O, and O=C-N of the monomer) [15,45]. At the same time, N1s spectra in Figure 4c of the Au@*p*DA-3-*p*MPTO chip comprised two characteristic peaks at 399.58 (-CO-NH- of TMAO;) and 403 eV (-N^+^(CH_3_)_2_-O-). As is shown in Figure 4d, the O1s spectrum of the *p*MPAO-grafted chip was divided into two peaks at 532.2 (O=C-N) and 531.0 eV (-(CH_3_)_2_N^+^-O-). These results demonstrated the presence of the characteristic molecular structure of *p*MPAO.

### 3.3. Adsorption of Ions on Zwitterionic Polymer

In this study, ion adsorption on *p*MPAO-grafted chips was investigated in the presence of different salts (NaCl, Na_2_SO_4_ and MgCl_2_) by monitoring the frequency and dissipation shifts (Δ*f* and Δ*D*). The results in Figure 5a–c showed that both Δ*f* and Δ*D* changed stepwise with salt concentrations on the initiator-coupled chip no matter which salt was applied. The higher salt concentration was, the higher value of −Δ*f* was obtained. It indicated an increased ion adsorption at higher salt concentrations. Likely, Δ*D* increased with an increasing salt concentration in Figure 5, revealing that ion adsorption induced the formation of a more flexible *p*PMAO layer on the chip surface. It was caused by the binding of hydrated ions with the polymer layer [34].

In this study, moreover, stepwise changes in Δ*f* and Δ*D* of initiator-coupled chips were also observed in Appendix A of the Appendix A. It was seen that there were larger values in −Δ*f* on the initiator-coupled chip than on the *p*MPAO-grafted chip at corresponding salt concentrations, exhibiting that the zwitterionic *p*DA layer had ion-strength dependence. The dependence was more significant in the presence of Na_2_SO_4_ and MgCl_2_. As Na_2_SO_4_ and MgCl_2_ were applied, more hydrated water was transferred into the coated layer via ion adsorption due to strong salt interaction with water [46], and a more flexible *p*DA layer formed. It is worth pointing out that the ion adsorption behavior of *p*MPAO brushes is distinct from that of PPC brushes and PSBMA brushes reported by Lin et al. [23]. It may be attributed to the stronger “anti-polyelectrolyte effect” of *p*MPAO brushes compared with PPC brushes and PSBMA brushes [47]. Because the distance between the anionic and cationic groups of monomers is shorter, pairing of TMAO-analog zwitterionic groups seems to occur more easily. Additionally, due to the proximity of O^−^ and N^+^ in the monomer, the electrostatic attraction between the cations and O^−^ of the *p*MPAO brush may be weakened [24]. As a result, lower ion mass was bound on the *p*MPAO-grafted chip at 1 and 10 mmol/L NaCl. With a further increase in salt concentration, −Δ*f* and Δ*D* on the *p*MPAO chip were comparable to those on initiator-coupled chip, revealing that the inner *p*DA layer may involve ion adsorption at 100 mmol/L NaCl. The influence of salt type to ion adsorption on the *p*MPAO-grafted chip (shown in Figure 5b,c) further confirmed the influence of co-existence of *p*MPAO and *p*DA to ion adsorption at the salt concentration of 100 mmol/L. The result of ion adsorption demonstrated that *p*MPAO grafting inhibited ion binding greatly.

The structural evolution of *p*MPAO brushes on the chip surface was further analyzed based on dissipation versus frequency profiles (*D*-*f* plots) in Figure 6. Due to a little change in frequency shift, the profile of Δ*D* against −∆*f* shrank into a spot at 1 mmol/L NaCl, and ellipse-like spots were observed at the bottom left of *D*-*f* plots. Therefore, two-stage characteristics were observed in *D*-*f* plots. It was seen in Figure 6a–c that Δ*D* increased with an increase in −∆*f* at a salt concentration of 10 mmol/L and the slopes were determined to be 1.02 × 10^−7^ in NaCl solution, 2.19 × 10^−7^ in Na_2_SO_4_ solution and 1.81 × 10^−7^ in MgCl_2_ solution. The tendency was also observed on initiator-coupled chips in Appendix A of the Appendix A. However, larger slopes of Δ*D/*−∆*f* on initiator-coupled chips were determined to be 4.7 × 10^−7^ in NaCl solution, 3.95 × 10^−7^ in Na_2_SO_4_ solution and 4.33 × 10^−7^ in MgCl_2_ solution at a salt concentration of 10 mmol/L. Smaller slopes of Δ*D/*−∆*f* on the *p*MPAO-grafted chip were consistent with the “anti-polyelectrolyte effect” mentioned above. With an increase in salt concentration, the *p*MPAO-grafted chip exhibited a different change in slope from the initiator-coupled chip at a salt concentration of 100 mmol/L. The result in Figure 6 showed that Δ*D* increased with larger slopes at 100 mmol/L, indicating that the *p*MPAO-grafted chip became more flexible at higher salt concentrations. It was consistent with the above statement that the inner *p*DA layer on the *p*MPAO-grafted chip may involve ion adsorption at 100 mmol/L NaCl. In contrast, slopes of Δ*D* against −∆*f* on the initiator-coupled chip tend to be smaller at 100 mmol/L (shown in Appendix A of the Appendix A), implying that the flexibility of the *p*DA layer on the initiator-coupled chip decreased. The result confirmed that a sequential *p*DA and *p*MPAO arrangement from chip surface on *p*MPAO-grafted chips exhibited different characteristics for ion adsorption from initiator-coupled chips with a simple *p*DA layer.

### 3.4. Adsorption of BSA on Zwitterionic pMPAO

In this study, BSA was applied to investigate the fouling behavior of *p*MPAO-grafted chips. The frequency and dissipation shift of BSA adsorption on the Au@*p*DA-2-*p*MPAO chip are shown in Figure 7. The result in Figure 7a showed that BSA adsorption on the initiator-coupled chip led to a decrease in Δ*f* to −22 Hz. As shown in Appendix A of the Appendix A, BSA adsorption on bare gold and *p*DA-coated chips had a similar frequency shift. However, *p*MPAO grafting led to a great decrease in Δ*f*, and the value of Δ*f* was determined to be −7.6 Hz as shown in Figure 7c. Based on Equation (1), the adsorption mass of BSA was determined to be 134.52 ng/cm^2^ on Au@*p*DA-2-*p*MPAO chip and 389.4 ng/cm^2^ on Au@*p*DA-2-Br chip. It revealed that the presence of *p*MPAO brush on the chip surface inhibited BSA adsorption greatly and led to a 65.5% reduction in BSA adsorption. Furthermore, it was seen in Figure 7a and c that Δ*D* had a different trajectory on the Au@*p*DA-2-*p*MPAO chip from the corresponding initiator-coupled chip and Δ*D* increased more gently on the *p*MPAO-grafted chip in the adsorption process. *D*-*f* plots in Figure 7 provide more information about the properties of binding, conformation, and viscoelasticity in the absorbed layer of proteins, and even the interaction of proteins–surfaces [48,49]. The result in Figure 7d further showed that Δ*D* increased with −Δ*f* at a larger slope of 4.98 × 10^−8^ on the Au@*p*DA-2-*p*MPAO chip. This value in slope was almost twice as high as that on the Au@*p*DA-2-Br chip (shown in Figure 7b). It is well known that protein adsorption always involves the net release of water molecules from the protein surface, leading to an increase in the rigidity of the adsorbed layer on the chip surface. Compared with the Au@*p*DA-2-Br chip, therefore, a larger slope on the Au@*p*DA-2-*p*MPAO chip suggested less net release of water molecules on the protein surface and *p*MPAO brush in protein adsorption. In practical terms, it signified much weaker BSA-*p*MPAO interaction and a weakly absorbed BSA on *p*MPAO brushes.

Figure 8 shows the frequency and dissipation shift of BSA adsorption on different *p*MPAO-grafted chips. The result in Figure 8a showed that there were smaller frequency shifts on the Au@*p*DA-3-*p*MPAO chip than on the Au@*p*DA-2-*p*MPAO chip. It revealed that a more significant inhibition of protein adsorption was attributed to the higher density of *p*MPAO brushes. However, an increase in DA concentration had no marked influence on the frequency shift to initiator-coupled chips (−18 Hz) in Appendix A of Appendix A. Based on Equation (1), the adsorption mass of BSA were calculated to be 292.25 ng/cm^2^ on the Au@*p*DA-3-Br chip and 61.95 ng/cm^2^ on the Au@*p*DA-3-*p*MPAO chip, indicating a 78% reduction in BSA adsorption on the Au@*p*DA-3-*p*MPAO chip compared with the Au@*p*DA-3-Br chip. However, the adsorption mass of BSA on the Au@*p*DA-4-*p*MPAO chip in Figure 8c could not be calculated because of overly fluctuating frequency data. Such fluctuation indicated no stable adsorption layer formed on the Au@*p*DA-4-*p*MPAO chip. It matched the evidence of a constant dissipation in protein adsorption as shown in Figure 8c. The *D*-*f* plots in Figure 8b further showed that Δ*D* increased with an increase in −Δ*f* on the Au@*p*DA-3-*p*MPAO chip and the slope was determined to be 5.37 × 10^−8^. This value was a little larger than the slope on the Au@*p*DA-2-*p*MPAO chip in Figure 8d. On the Au@*p*DA-4-*p*MPAO chip, furthermore, the dissipation value remained stable in BSA adsorption, indicating that there was no change in water molecule in the *p*MPAO brush. It exhibited different *D*-*f* characteristics from Au@*p*DA-2-*p*MPAO and Au@*p*DA-3-*p*MPAO chips, signifying that BSA-*p*MPAO interaction could be ignored. It implied that BSA cannot bind to the *p*MPAO brush. It means that the *p*DA-4-*p*MPAO surface is nearly a non-fouling interface, consistent with reports of non-fouling materials. We first visually evaluated the interaction between BSA and *p*MPAO brush using QCM-D. Therefore, the best resistance to BSA adsorption was obtained on the Au@*p*DA-4-*p*MPAO chip.

## 4. Conclusions

In this study, an innovative TMAO-analog monomer, MADMPAO, was synthesized by oxidation reaction and grafted onto the QCM chips by the combination of bio-inspired *p*DA and SI-ATRP technology. The results showed that *p*MPAO grafting brought about a great increase in the thickness of the polymer layer, and a decrease in contact angles, and a change in roughness, demonstrating successful *p*MPAO grafting on *p*DA-coated chips. Ion adsorption on Au@*p*DA-*p*MPAO chip is distinct from that on *p*DA-coated chips, and ion-strength dependence on *p*DA-coated chips was inhibited by *p*MPAO grafting, especially at low salt concentrations. It is attributed to a stronger ‘anti-polyelectrolyte effect’ for *p*MPAO brushes. Therefore, a sequential *p*DA and *p*MPAO arrangement from the chip surface provided different characteristics for ion adsorption from a simple *p*DA layer. Furthermore, BSA adsorption was inhibited greatly on *p*MPAO-grafted chips characterized by a much smaller value of −Δ*f*. Compared with the corresponding initiator-coupled chips, larger slopes of Au@*p*DA-2-*p*MPAO and Au@*p*DA-3-*p*MPAO chips suggested less net release of water molecules in protein adsorption, signifying weaker BSA-*p*MPAO interaction. In QCM analysis, the Au@*p*DA-4-*p*MPAO chip kept stable dissipation in BSA adsorption, indicating that there was no change in water molecules in the *p*MPAO brush. It indicated that BSA-*p*MPAO interaction could be ignored, and the Au@*p*DA-4-*p*MPAO chip exhibited the best resistance to BSA adsorption. The research provided a novel monomer for zwitterionic polymer and exhibited antifouling characteristics of *p*MPAO brush on chip surface.

## Figures and Tables

**Figure 1 polymers-16-01634-f001:**
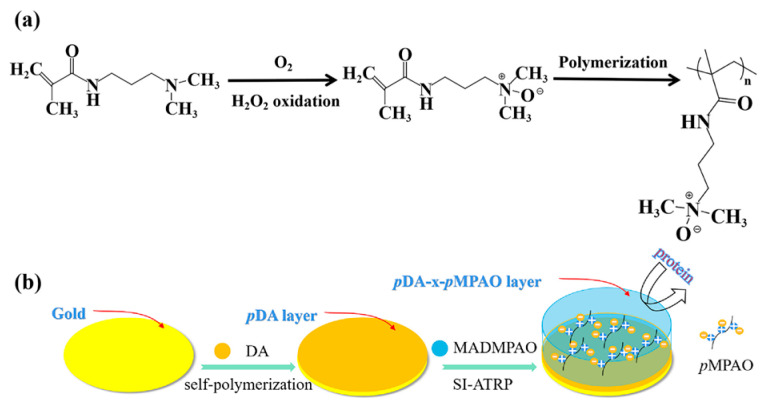
Synthesis of *p*MPAO-grafted chips via SI-ATRP. (**a**) Synthesis reaction of MAPMPAO monomer and *p*MPAO. (**b**) Synthesis of *p*MPAO-grafted chips via SI-ATRP.

**Figure 2 polymers-16-01634-f002:**
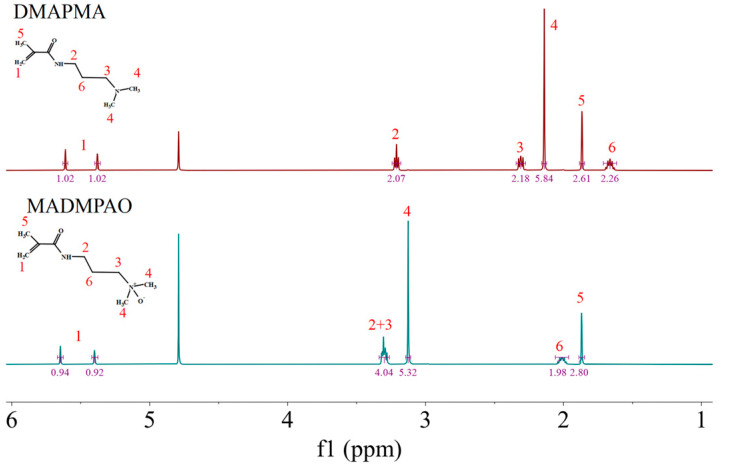
H NMR spectrum of DMAPMA and MADMPAO in D_2_O.

**Figure 3 polymers-16-01634-f003:**
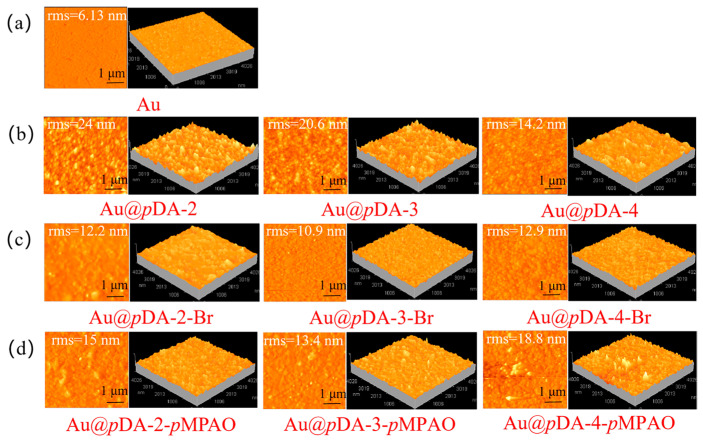
2D and 3D AFM images of bare gold (**a**), *p*DA-coated (**b**), initiator-coupled (**c**) and *p*MPAO-grafted chips (**d**). *p*DA coating on bare gold chip were prepared from the CuSO_4_/H_2_O_2_-triggered rapid deposition. The dimension of each scan image is 5 × 5 μm^2^.

**Figure 4 polymers-16-01634-f004:**
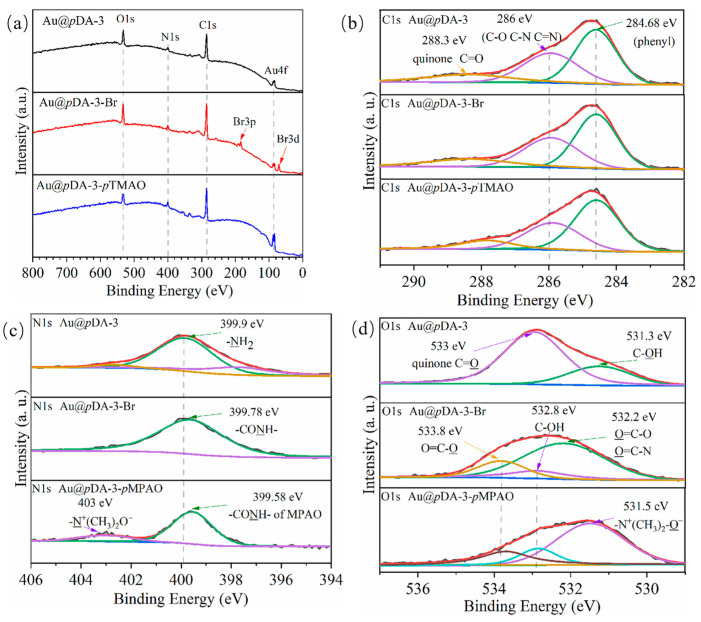
XPS spectra for Au@*p*DA-3, Au@*p*DA-3-Br and Au@*p*DA-3-*p*MPAO chips. (**a**) Full spectrum, (**b**) C1s spectrum, (**c**) N1s spectrum and (**d**) O1s spectrum.

**Figure 5 polymers-16-01634-f005:**
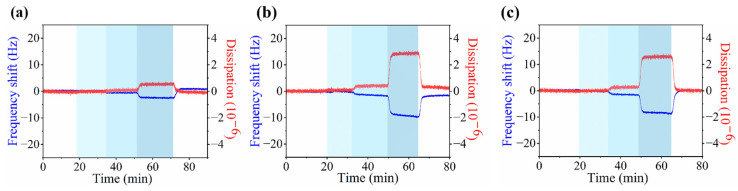
Monitoring of frequency and dissipation shift versus time on *p*MPAO-grafted chips in the presence of NaCl (**a**), Na_2_SO_4_ (**b**) and MgCl_2_ (**c**) with different concentrations (1.0, 10, and 100 mmol/L).

**Figure 6 polymers-16-01634-f006:**
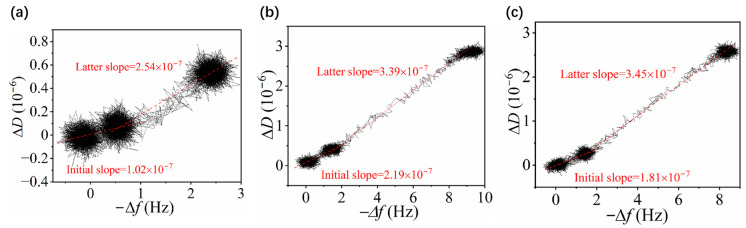
Monitoring of dissipation shift versus frequency on *p*MPAO-grafted chips in the presence of NaCl (**a**), Na_2_SO_4_ (**b**) and MgCl_2_ (**c**) with different concentrations (1.0, 10, and 100 mmol/L).

**Figure 7 polymers-16-01634-f007:**
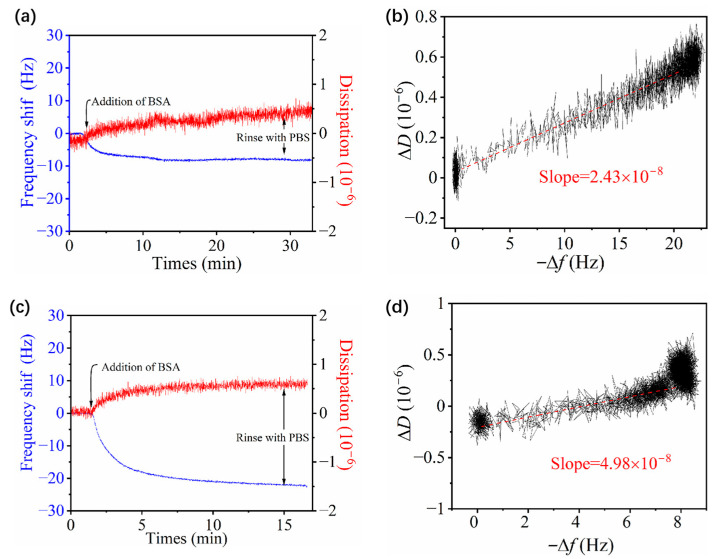
QCM measurement of BSA adsorption on Au@*p*DA-2-Br and Au@*p*DA-2-*p*MPAO chips. (**a**) Time-resolved profile of Δ*f* and Δ*D* on Au@*p*DA-2-Br chips, (**b**) *D*-*f* plot on Au@*p*DA-2-Br chip, (**c**) time-resolved profile of Δ*f* and Δ*D* on Au@*p*DA-2-*p*MPAO chip and (**d**) *D*-*f* plot on Au@*p*DA-2-*p*MPAO chip.

**Figure 8 polymers-16-01634-f008:**
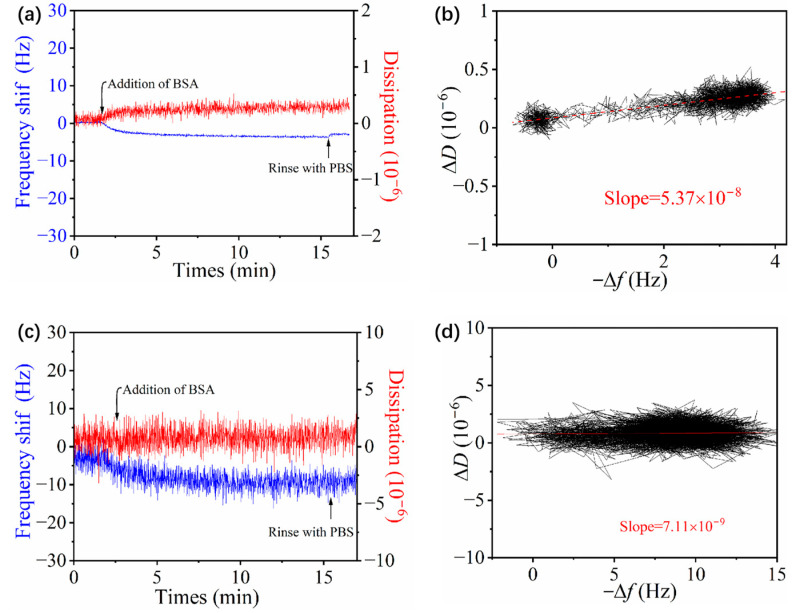
QCM measurement of BSA adsorption on Au@*p*DA-3-*p*MPAO and Au@*p*DA-4-*p*MPAO chips. (**a**) Time-resolved profile of Δ*f* and Δ*D* on Au@*p*DA-3-*p*MPAO chip, (**b**) *D*-*f* plot on Au@*p*DA-3-*p*MPAO chip, (**c**) time-resolved profile of Δ*f* and Δ*D* on Au@*p*DA-4- *p*MPAO chip and (**d**) *D*-*f* plot on Au@*p*DA-4-*p*MPAO chip.

**Table 1 polymers-16-01634-t001:** Element composition of surfaces of representative DA-coated, initiator-coupled and *p*MPAO-grafted chips analyzed by XPS.

Chips	XPS Surface Composition (at %)			
	C1s	N1s	O1s	Br3d
Au@*p*DA-3	73.44	8.65	17.91	0
Au@*p*DA-3-Br	69.8	7.9	18.61	3.69
Au@*p*DA-3-*p*MPAO	72.61	8.05	19.34	0

## Data Availability

Data will be made available on request.

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
