# Peer review of "Bioinspired Dopamine and N-Oxide-Based Zwitterionic Polymer Brushes for Fouling Resistance Surfaces"

_polymers, 2024, doi:10.3390/polym16121634_

Round 1

Reviewer 1 Report

Comments and Suggestions for Authors

Publish after major revisions noted below:

1.       Enhance the quality of the manuscript.

2.       Further clarify the originality and innovative aspects of the paper.

3.       Ensure that the full name of QCM (Quartz Crystal Microbalance) is explicitly stated in the abstract.

4.       Enhance the logical flow of the introduction.

5.       Ensure that in Figure 1b, chemical structures are properly labeled. Present chemistry (b) before illustration (a).

6.       Clearly label the purity of raw materials.

7.       Substitute 'new' with a different word or phrase.

8.       Conduct a comparative analysis of the impact of the author's studies against other anti-fouling strategies documented in the literature.

9.       Reorganize the conclusion, highlighting the most significant summary points.

Comments on the Quality of English Language

Enhance the quality of the manuscript.

Author Response

Responses to Reviewers’ Comments

Dear Editor,

We sincerely thank the editor and all reviewers for their valuable comments to improve the quality of our manuscript. According to your nice comments, we have made extensive corrections to our original manuscript, and the detailed response to all the coments are listed below. We hope that the revised manuscript could meet the requirement of the journal. Thank you in advance.

Reviewer #1

  1. Enhance the quality of the manuscript.

Response: Thanks for the comment. The manuscript was revised carefully based on the suggestions and comments.

  1. Further clarify the originality and innovative aspects of the paper.

Response: Thanks for the useful comment. We revised the introduction section and emphasized the originality and innovation of research. In this work, a novel TMAO-analog monomer was synthesized, and pTMAO grafted chips were fabricated and characterized. To the best of our knowledge, moreover, it is the first time that the ions and protein adsorption behavior on pMPAO brushes on pDA-coated chip are investigated by QCM-D.

  1. Ensure that the full name of QCM (Quartz Crystal Microbalance) is explicitly stated in the abstract.

Response: Thanks for the careful comment. In the revised manuscript, the full name of QCM was added (lines 13-14).

  1. Enhance the logical flow of the introduction.

Response: Thanks for the comment. The background was reorganized in the revised manuscript. In the introduction, we added signal words and concluding sentences and removed some redundant sentences.

  1. Ensure that in Figure 1b, chemical structures are properly labeled. Present chemistry (b) before illustration (a).

Response: Thanks for the careful comment. We redrew Figure 1, and the synthesis of the monomer and polymerization was reorganized in Figure 1a. Moreover, the chemical structure of monomer and polymer was checked carefully to make sure the structure was properly labeled.

  1. Clearly label the purity of raw materials.

Response: Thanks for the comment. In section 2.1, the purity of raw materials were added (lines 112-118).

  1. Substitute 'new' with a different word or phrase.

Response: Thanks for the comment. The words of ‘original’ and ‘inventive’ were used for substituting 'new' (lines 95, 124, 205, 223).

  1. Conduct a comparative analysis of the impact of the author's studies against other anti-fouling strategies documented in the literature.

Response: Thanks for the comment. In the revised manuscript, the result in this work was further discussed in Section 3.3 (Line 307~314) and Section 3.4 (Line 399~402) based on previous reports from other groups (refs. [23], [24] and [47]).

  1. Reorganize the conclusion, highlighting the most significant summary points.

Response: Thanks for the comment. We reorganized the conclusion, and the significant point was highlighted.

Reviewer #2

  1. Determine the purity of compounds

Response: Thanks for the comment. In this research, an original monomer was synthesized and the results of 1H NMR and HRMS indicated that no residual reactant and other impurity in the final product. We had wanted to determine the purity of this monomer, but there were no standards to use as a control. In ongoing work, we attempt to develop direct methods for the determination of monomer purity.

  1. Provide 13C NMR spectrum

Response: Thanks for the comment. As a complement to 1H NMR, 13C NMR can provide more comprehensive structural information. In my opinion, the result of 1H NMR and HRMS have been able to preliminarily confirm the structure of the monomer. In a previous report by Zhang et al., 1H NMR was also applied to characterize TMAO polymer.

  1. In this article, a new monomer was synthesized. However, its structure has not been sufficiently confirmed. So the NMR spectrum of tmao monomer is not shown.

Response: Thanks for the comment. As reported by Zhang et al., the 1H NMR of PDMAEMA-b-PS copolymer, one of the TMAO polymers, was shown. In this paper, the 1H NMR spectrum of the substrate (DMAPMA) and the monomer (MADMPAO) was shown in Figure 2. TMAO is a general term for a class of compounds with N- and O- groups, and MADMPAO is one of the TMAO-analog monomers.

  1. When developing polymer analogues, it is necessary to consider a set of properties that are necessary for the successful operation of polymers, namely: glass transition temperature, strength, critical deformation, level of adhesion to a specific material to which it is applied. Such studies are not provided in the article. Moreover, it has not been proven that the polymer has completely cured (99-100% conversion rate).

In my opinion, the conclusion that an analogue has been developed is not correct without these results.

Response: Thanks for the comment. In this work, the monomer was synthesized  and grafted onto the chip surface. Straight to the point, the novelty of this paper is that the QCM-D was used for the first time to study the ion adsorption and BSA adsorption behavior on the  pDA-pTMAO coating.

  1. 14. Figure 3 is repeated 6 times in the text of the article.

Response: Thanks for the comment. In original manuscript, the letters of a to d in Figure 3 were not be bold, and it led to misunderstanding by readers. In the revision, the manuscript was updated in combination with the next comment.

  1. 15. In Section 3.2, do not use bold font.

Response: Thanks for the comment. As same as the comment 14, all the bold font related to figures were replaced by regular font.

  1. 16. Figure 4. It is necessary to design the axes and their captions in the same size font.

Response: Thanks for the comment. In the revised manuscript, Figure 4 was updated.

  1. 17. Figures 5, 6, 7, 8 are practically unreadable. Increase the font size of the axes, captions for the axes and for the figures themselves.

Response: Thanks for the comment. The axes and font size of the title of a single graph had increased to 36. This font size is enough to be readable.

  1. 18. References:

- Double-check the abbreviated names of journals in the references, for example, [14]

- Check DOI in references [10], [34]

Response: Thanks for the comment. In the revised manuscript, we updated the references.

  1. It is worth noting that the results presented in the article are interesting, but refinement is required, after which it will be possible to recommend the article for publication. In my opinion, for publication in the journal «Polymers», it is necessary to study the polymer in more detail.

Response: Thanks for the great comment. It is precisely because pTMAO is grafted onto the chip that interesting results can be obtained, and our interest focuses on the interaction behavior between polymers and molecules. In this work, the detailed study about ion and protein adsorption on pMPAO brushes were investigated. The result exhibited the best resistance properties for protein adsorption. It provided an insight into anti-fouling characteristics of zwitterionic polymer based on the novel monomer.

Reviewer 2 Report

Comments and Suggestions for Authors

1. Determine the purity of compounds

2. Provide 13C NMR spectrum

3. In this article, a new monomer was synthesized. However, its structure has not been sufficiently confirmed. So the NMR spectrum of tmao monomera is not shown.

4. When developing polymer analogues, it is necessary to consider a set of properties that are necessary for the successful operation of polymers, namely: glass transition temperature, strength, critical deformation, level of adhesion to a specific material to which it is applied. Such studies are not provided in the article. Moreover, it has not been proven that the polymer has completely cured (99-100% conversion rate).

In my opinion, the conclusion that an analogue has been developed is not correct without these results.

5. Figure 3 is repeated 6 times in the text of the article.

6. In Section 3.2, do not use bold font.

7. Figure 4. It is necessary to design the axes and their captions in the same size font.

8. Figures 5, 6, 7, 8 are practically unreadable. Increase the font size of the axes, captions for the axes and for the figures themselves.

9. References:

- Double-check the abbreviated names of journals in the references, for example, [14]

- Check DOI in references [10], [34]

It is worth noting that the results presented in the article are interesting, but refinement is required, after which it will be possible to recommend the article for publication. In my opinion, for publication in the journal «Polymers», it is necessary to study the polymer in more detail.

Author Response

(The authors gave the same response as above.)

Round 2

Reviewer 1 Report

Comments and Suggestions for Authors

The author revised the manuscript based on comments, the paper is good to publish.

Author Response

Dear Editor,

We sincerely thank the editor and all reviewers for their valuable feedback once again. We answer the following main questions and hope to clarify them. At the same time, updated conclusion section was highlighted in yellow color.

Reviewer #1

Response: Thanks for the help and valuable suggestion.

Reviewer 2 Report

Comments and Suggestions for Authors

The authors have revised the article. In my opinion, the conclusions are not entirely justified. The authors did not answer some questions or ignored them. No answer was received regarding the degree of curing of the polymer. The fact is that if the polymer is under-cured, then over time it will still cure and its properties, which are described by the authors, will change. The monomer structure has not been confirmed. This journal is primarily about polymers. I would recommend another journal for this research. It is at the discretion of the journal editor whether to accept this article or not.

Author Response

Dear Editor,

We sincerely thank the editor and all reviewers for their valuable feedback once again. We answer the following main questions and hope to clarify them. At the same time, updated conclusion section was highlighted in yellow color.

Reviewer #2:

  1. The authors have revised the article. In my opinion, the conclusions are not entirely justified. The authors did not answer some questions or ignored them. No answer was received regarding the degree of curing of the polymer. The fact is that if the polymer is under-cured, then over time it will still cure and its properties, which are described by the authors, will change.

Response: Thanks for the comment. In the revised manuscript, the conclusion was concentrated to highlight the focal point in this research. As indicated by our reviewer, curing method and process had a great influence in the properties of polymer layers. In this study, furthermore, the synthesis of pMPAO-grafted chip did not involve in the chip treatment by any specific curing methods, such as thermal heating and radiation heating. SI-ATRPAs was applied to synthesize pMPAO brushes in the absence of oxygen and in the presence of catalysts. As described in Section 2.4 (Line 171-173), the chips were removed from the flasks and washed thoroughly and used for characterization and QCM measurement directly. In this case, the change in polymer properties could be ignored. It was also verified by repeated QCM measurement. In this study, all QCM measurements were carried out triplicate as described in Lines 189-190.

  1. The monomer structure has not been confirmed.

Response: Thanks for the comment. In this study, monomer structure is confirmed by 1H NMR. 1H NMR has extensively been applied in the characterization of TMAO and derivatives [Journal of Membrane Science, 2021, 640: 119855; Macromolecules, 2021, 54: 4236−4245]. In this study, the monomer structure was verified by 1H NMR and HRMS. 1H NMR of DMAPMA (reactant) and MADMPAO (product) are shown in Figure 2. 1H NMR spectra of MADMPAO could be in good agreement with the predicted spectrum obtained from MestReNova software (as shown in Figure R1 below). Furthermore, HRMS (mass/charge ratio): calculated for C9H18N2O2, 187.1368 ([M+H]+); found 187.1442 (lines 216-218). NMR and HRMS results confirmed a successful synthesis of MADMPAO.

Figure R1, pls see attached file.

Figure R1  Predicted 1H NMR spectrum of MADMPAO by MestReNova
